# A Multi-Pronged Technique for Identifying *Equisetum palustre* and *Equisetum arvense*—Combining HPTLC, HPLC-ESI-MS/MS and Optimized DNA Barcoding Techniques

**DOI:** 10.3390/plants11192562

**Published:** 2022-09-28

**Authors:** Afoke Ibi, Min Du, Till Beuerle, Dennis Melchert, Julia Solnier, Chuck Chang

**Affiliations:** 1ISURA, 101-3680 Bonneville Place, Burnaby, BC V3N 4T5, Canada; 2Institute of Pharmaceutical Biology, Faculty of Life Sciences, Technical University of Braunschweig, 38106 Braunschweig, Germany

**Keywords:** *Equisetum arvense*, *Equisetum palustre*, *ITS*, internal transcribed spacer, alkaloids, palustrine, palustridiene, HPTLC, LCMS

## Abstract

The most prominent horsetail species, *Equisetum arvense*, has an array of different medicinal properties, thus the proper authentication and differentiation of the plant from the more toxic *Equisetum palustre* is important. This study sought to identify different samples of *E. arvense* and *E. palustre* using three analytical methods. The first method involved the use of HPTLC analysis, as proposed by the European Pharmacopoeia. The second, HPLC-ESI-MS/MS, is capable of both identification and quantification and was used to determine the *Equisetum* alkaloid content in each sample. A third method was DNA barcoding, which identifies the samples based on their genetic make-up. Both HPTLC and HPLC-ESI-MS/MS proved to be suitable methods of identification, with HPLC-ESI-MS/MS proving the more sophisticated method for the quantification of alkaloids in the *Equisetum* samples and for determining the adulteration of *E. arvense*. For DNA barcoding, optimal primer pairs were elucidated to allow for the combined use of the *rbcL* and *ITS* markers to accurately identify each species. As new DNA marker sequences were added to GenBank, the reference library has been enriched for future work with these horsetail species.

## 1. Introduction

The horsetail herb (*Equisetum arvense*) is a species of the genus *Equisetum* (family: *Equisetaceae*) most commonly found in the Northern Hemisphere, with other species scattered throughout select areas of Africa, Asia, and South America [1]. Of the 15 species that constitute this genus, only *E. arvense* has been approved for use as an herbal remedy [2]. Its medicinal properties include antioxidant, anti-inflammatory, diuretic, dermatological, and immunological properties, to name a few [3]. *E. arvense* contains many different compounds such as minerals, flavonoids, phenolic glycosides, alkaloids, and triterpenoids [4]. This herb’s wide use in herbal remedies necessitates proper authentication during the production process. The need for accurate identification is of even greater importance when one considers the potential for the *E. arvense* species to be adulterated with the more toxic *Equisetum palustre*. While all horsetail species contain trace levels of nicotine [5], *E. palustre* is known to have higher levels of spermidine derived piperidine alkaloids (up to 0.1%), and it is assumed that a content of 5% of *E. palustre* in animal feed may have poisonous effects [6]. Though the use of herbal supplements has increased in popularity worldwide, they remain unregulated in some countries [7]. Fortunately, tests that confirm the identity and quality of these products can be found in monographs of international pharmacopoeias. One such monograph is from the European Pharmacopoeia, where tests such as macroscopy and microscopy as well as thin layer chromatography (TLC) for identification have been described [8]. However, since most herbal supplements are sold in the form of powders, capsules, tablets, or tinctures, macroscopy and microscopy methods cannot be applied to such samples, which leaves TLC as the primary method for the identification of horsetail herbal supplements.

The TLC test in this monograph can be employed to determine adulteration by detecting the presence of foreign materials, in addition to providing the identification of *E. arvense*. To verify the authenticity of the sample herbal material, the fingerprint of the sample chromatogram may be compared to that of an authenticated reference sample using criteria specified in the monograph. However, while TLC fingerprint analysis is frequently used as a technique for identification because it is straightforward and inexpensive, this methodology has a number of drawbacks. For one, chemical markers of a plant can vary depending on its growth and harvesting conditions [9]. Additionally, the same chemical markers might be found in other species. Finally, herbal materials may sometimes be adulterated with synthetic substances to mimic the expected fingerprint [10]. With these factors in mind, it is easy to see that depending on TLC alone may produce false positive identification results. Further identification methods are thus advised. For example, focusing on the quantification of the alkaloid content in horsetail samples could aid in determining adulteration and help prevent the use of toxic herbal materials. Such analysis has been achieved using high-performance liquid chromatography electrospray ionization tandem mass spectrometry (HPLC-ESI-MS/MS). The major *Equisetum* type-alkaloids present in *E. palustre* are two piperidine alkaloids: palustrine and palustridiene (Figure 1). They are present in comparatively high levels, ranging from 120 to 883 mg/kg [5]. *E. arvense,* on the other hand, does not contain *Equisetum* type-alkaloids [11]. Because of this, it is possible to categorically rule out *E. arvense* when a sample is confirmed to contain these alkaloids. However, the converse does not hold true. The general absence of *Equisetum* alkaloids does not conclusively prove that the sample is *E. arvense*, because other *Equisetum* species have also been shown to lack these alkaloids [12]. Therefore, alkaloid analysis using HPLC-ESI-MS/MS alone cannot always distinguish one species from another.

Another method that can be used to identify herbal materials is DNA barcoding. Following its success in use with animals, DNA barcoding has been proposed as a method to accurately identify plant species since the early parts of this century [14,15,16,17,18,19]. Currently, complete genomic sequencing is expensive and time-consuming, and would require extensive data reduction to determine inter-species differences. Therefore, rather than determining the complete genome of each sample, specific marker sequences within the target plant species are employed for identification purposes. These sequences have been empirically determined to provide sufficient variation between species and, yet, conservative enough that little or no variation is observed between members of the same species. By querying the marker sequences of an unknown sample in a reference database such as GenBank with algorithms such as the Basic Local Alignment Search Tool (BLAST), the identity of the unknown sample can be determined if it matches existing sequences in the database. This method of identification is robust and can be accurate to the species level, especially when several markers are examined in combination. Examples of marker sequences that have been used for this purpose in plants include *ITS*, *aptF*-H, *matK*, *pasK*, *rbcL*, *rpoB*, *rpoC1*, *trnH*, and *trnL*. The markers *rbcL* and *ITS* are among the most frequently used barcoding regions for plant identification [14,15,20,21] and have been used by other research scientists in a previous study of horsetail [2]. The success of DNA barcoding in identifying herbal material, however, is dependent on the data available in the relied upon reference database. Understandably, less studied species typically have less sequence data in reference databases.

The current study compares the data of known and unknown *Equisetum* samples in powder form. High-performance thin layer chromatography (HPTLC), HPLC-ESI-MS/MS, and DNA barcoding were used to accurately identify samples of *E. arvense* and distinguish them from *E. palustre*.

## 2. Results

### 2.1. HPTLC

The results of the HPTLC analysis are shown in Figure 2 and Figure 3 where a comparison of several *E. arvense* and *E. palustre* methanol extracts are illustrated. In both figures, the chemical standards rutin, hyperoside and caffeic acid are present in ascending order on track 1. Figure 2 shows the *E. arvense* chromatogram, which includes the *E. arvense* sample extracts A1 to A9 on tracks 4 to 12. The TLC identification of *E. arvense* by the European Pharmacopoeia indicates two blue zones between the reference hyperoside and rutin, an orange zone just above hyperoside, two greenish-blue zones between hyperoside and caffeic acid, and two red zones right below the solvent front. The European Pharmacopoeia also indicates potential adulteration in the chromatograms of *E. arvense* where there are two greenish zones just above the application line that are 5% more intense than the corresponding greenish zones in the chromatogram for *E. palustre*.

Except for samples A2, A8, and A9, all of the *E. arvense* horsetail samples conformed to the indicated acceptance criteria with no signs of adulteration by *E. palustre*. Samples A6 and A7 each displayed a single green band near the application line but the fact that they were both low intensity and singular bands as opposed to double bands excused them from being categorized as adulterated samples. Samples A2 and A8 presented all the bands indicated by the acceptance criteria except for a blue zone just above hyperoside instead of the expected orange zone. While sample A9 also shared this characteristic blue instead of orange zone just above hyperoside, its chromatogram was still distinctly different from all of the other *E. arvense* samples due to the absence of not just the two greenish zones below caffeic acid, but also other secondary bands. It is of interest to note that all three of these non-conforming *E. arvense* samples were sourced from Canada. This indicates that a variation in phytochemistry, potentially due to varying local conditions, might affect the positive HPTLC-based identification of *E. arvense* using the criteria outlined by the European Pharmacopoeia.

On the other hand, despite their differing sources and environments, the *E. palustre* chromatograms displayed in Figure 3 appear to be more consistently similar. Where significant variation was found, it was limited to the variable band intensities rather than the discovery of additional bands.

### 2.2. HPLC-ESI-MS/MS

An external calibration method was used to quantify the palustriene and palustridiene. The investigated linear calibration ranged from 0.625 to 40 ng/mL by using authentic reference material. Analytical results were calculated using the original sample weight, the slopes of the corresponding calibration curve, the area-values of the analytes, and the appropriate dilution factors of the corresponding sample preparation, as shown in Equation (1). A sample chromatogram with the calibration curve is shown in Figure 4.
(1)$$alkaloid\;content\left(\frac{ng}{mg}\right)=\frac{\frac{A_{alkaloid}\cdot df}{S}}{m_{sample}}$$

*A_alkaloid_*: area under the peak for the MS/MS-transition of the quantifier ion of for the corresponding alkaloid*df*: dilution factor*m_sample_*: sample weight*s*: slope of the corresponding calibration curve for the alkaloid

*E. palustre* is usually distinguished by a high concentration of *Equisetum* alkaloids such as palustrine and palustridiene. *E. arvense*, on the other hand, is distinguished by its absence of measurable *Equisetum* alkaloids. As seen in Table 1, the *E. arvense* samples had no quantifiable amounts of alkaloids with the exception of sample A1, which contained 6.2 µg/g of palustrine. As this is atypical for *E. arvense*, and given the small amount of palustrine present, it is plausible to assume that this sample material was contaminated by an *E. palustre* sample.

The amount of *Equisetum* alkaloids present in all ten samples (Table 2) was well within the known range of *Equisetum* alkaloid levels, as derived from past studies of *E. palustre* [6]. Variations in *Equisetum*-type alkaloid contents (sum of palustrine and palustridiene) can be explained by the ontogenetic stage of development, temperature, site, and environmental conditions [5,6,13].

### 2.3. DNA Barcoding

It was found that the DNA samples produced quality results most consistently when diluted 10-fold. When the samples were undiluted, amplification would frequently fail. This was especially true for the *E. arvense*-containing samples. This suggests the potential presence of PCR inhibitors in the *E. arvense* plant, which can be eluted along with DNA.

#### 2.3.1. *rbcL* Marker

The *rbcL*-a primer pair generated excellent quality sequences, generally exceeding 600 base pairs in length. This marker sequence was used to correctly identify P1, P2, P3, P4, P5, P6, P8, P9, P10, A2, A6, A7, A8, and A9 as their anticipated horsetail species (Appendix A Appendix A), but unfortunately failed to amplify in P7, A1, A3 and A5. As expected, different species within the genus *Equisetum* could be matched when queried using BLAST. For the “P” samples, *E.*
*palustre* consistently appeared as the best match (highest “Max Score”, 100% Query Cover, lowest E value and highest Percent Identity) (Table 3; Appendix A Appendix A). In a similar manner, for “A” samples, *E. arvense* was always the best match (Table 3; Appendix A Appendix A). For sample A4, the identification was to the genus level (*Equisetum* spp.), as several different *Equisetum* species were matched with equal likelihood (Table 3; Appendix A Appendix A). Overall, the *rbcL* marker, specifically using the *rbcL*-a primer pair, reliably distinguished between *E. palustre* and *E. arvense*. Insufficiencies seen with amplification could be attributed to the DNA quality or the presence of PCR inhibitors.

#### 2.3.2. *ITS* Marker

For the *ITS* marker, three different forward and reverse primer combinations, referenced from the work of Cheng et al. [21], were tried. The pairs were: ITS-S2F + ITS4, ITS-p3 + ITS-u4, and ITS-p5 + ITS-u2. It was discovered that the ITS-p3 and ITS-u4 pair (herein referred to as p3u4) consistently generated good quality sequence data and identified species within the *Equisetum* genus.

For *E. palustre*, vouchered plants from Germany (P3–P10) generated sequences between 464 to 473 base pairs in length when using p3u4; reference samples from commercial sources produced shorter sequences at 452 base pairs (P2, from ChromaDex) and 123 base pairs (P1, from European Pharmacopoeia).

For *E. arvense*, vouchered plants from Germany (A3–A7) generated p3u4 sequences between 306 to 315 base pairs; vouchered plants from British Columbia, Canada (A8–A9) resulted in 311–314 base pairs. Sample A1 failed to amplify, while sample A2 amplified poorly, with only 95 base pairs.

Even when the *rbcL* marker was unsuccessful at identifying *Equisetum*, at least to the genus level, as in the cases of P7, A3, and A5, their p3u4 results matched to the *Equisetum* genus. This suggests a level of robustness of the p3u4-derived *ITS* marker on par with or even exceeding that of *rbcL*. In order to identify these three samples to the species level, the Multiple Sequence Comparison by Log-Expectation (MUSCLE) tool was used. With MUSCLE, we aligned each of P7, A3, and A5 with the rest of the vouchered samples and discovered that P7 aligned well with all of the *E. palustre* samples, while A3 and A5 aligned best with all of the *E. arvense* samples. Such an alignment is, in fact, the basis of DNA barcoding. Since we had empirical data from vouchered plants, the identities of which were confirmed at the species level, it allowed us to verify the identities of P7, A3 and A5 as *E. palustre*, *E. arvense*, and *E. arvense*, respectively.

An interesting pattern emerged when we evaluated the p3u4 results: despite good quality DNA sequences and identifications within the *Equisetum* genus, the p3u4 data never specifically matched neither *E. palustre* nor *E. arvense* to the species level when queried using BLAST. This prompted us to re-examine the reference data available for BLAST. Since the BLAST program compares against the GenBank library, sequences were retrieved from existing *ITS* accessions in the GenBank database and queried with BLAST to evaluate the outcome. As of 2 June 2022, no *E. palustre ITS* accessions were found in GenBank. At the same time, eight *E. arvense ITS* accessions existed in GenBank:OL304248.1 (*Equisetum arvense*)OL304249.1 (*Equisetum arvense*)OL304250.1 (*Equisetum arvense*)MW040322.1 (*Equisetum arvense*)MW040323.1 (*Equisetum arvense*)MW040324.1 (*Equisetum arvense*)Y11470.1 (*Equisetum arvense*)Y11471.1 (*Equisetum arvense*)

Sequences from the eight accessions were downloaded, and then each was queried in BLAST. Each of these accessions was expected to first, identify itself as the best match and second, match the *ITS* accessions from other species within the *Equisetum* genus, with varying degrees of likelihood. It was observed that OL304248.1, OL304249.1, OL304250.1, MW040322.1, MW040323.1, and MW040324.1 did match themselves and each other (Appendix A Appendix A); however, they did not match any other *Equisetum* species. Unexpectedly, they consistently matched green algae, *Chloroidium* spp. and *Chlorella* spp., with *Chloroidium* spp. as the top match. In some cases, as seen with OL304249.1 and OL304250.1, the green algae accession’s E-value was only preceded by the queried accession itself, indicating a strong match (1 × 10^−169^ for *Chloroidium* spp. compared to 7 × 10^−173^ for OL304249.1, and 3 × 10^−171^ for *Chloroidium* spp. compared to 7 × 10^−173^ for OL304250.1; Appendix A Appendix A). However, in other cases, as seen with MW040322.1, MW040323.1, and MW040324.1, its E-value was indistinguishable from the queried accession (1 × 10^−169^ for both *Chloroidium* spp. and MW040322.1, 1 × 10^−170^ for both *Chloroidium* spp. and MW040323.1, 2 × 10^−168^ for both *Chloroidium* spp. and MW040324.1; Appendix A). Furthermore, Y11470.1 only matched itself and nothing else (Appendix A Appendix A), as did Y11471.1 (Appendix A Appendix A). The lack of matches to other *Equisetum* species was unusual, as was the identification with green algae species.

The process was next repeated with *ITS* accessions from other *Equisetum* species in GenBank (accessed on 5 January 2022):DQ377154.1 (*Equisetum telmateia* subsp. *telmateia*) (Appendix A Appendix A)DQ377155.1 (*Equisetum variegatum* subsp. *variegatum*) (Appendix A Appendix A)DQ377156.1 (*Equisetum × meridionale*) (Appendix A Appendix A)DQ377157.1 (*Equisetum × moorei*) (Appendix A Appendix A)AF448794.1 (*Equisetum ramosissimum*) (Appendix A Appendix A)KT960212.1 (*Equisetum variegatum*) (Appendix A Appendix A)EU328339.1 (*Equisetum hyemale* subsp. *affine*) (Appendix A Appendix A)MT198943.1 (*Equisetum* environmental sample clone EDNA16-0043490) (Appendix A Appendix A)

The results from the above accessions did adhere to our expectations. Each identified itself as the best match, followed closely by other *Equisetum* species. Non-*Equisetum* species were sometimes identified but were nowhere near the top matches. These observations differed from what was seen with the eight *E. arvense ITS* accessions. The only exception was KT960212.1 (*Equisetum variegatum*), which identified a single instance of *Equisetum* (itself), while all other BLAST hits belonged to *Trebouxia* (green algae) (Appendix A Appendix A). BLAST was also used to align each *E. arvense ITS* accession with the other *Equisetum* accessions listed above to manually examine their similarities. None of the *E. arvense* accessions produced a significant similarity with the other *Equisetum* species. Furthermore, using MUSCLE, we aligned the eight GenBank *E. arvense ITS* sequences with ITS-p3u4 sequences from our vouchered samples (A3–A9) (Appendix A Appendix A). The alignment showed a clear disparity between GenBank sequences and our sequences. As there were no *E. palustre ITS* accessions in GenBank, we were unable to repeat the process with *E. palustre* sequences.

## 3. Discussion

The results from all three identification methods (HPTLC, HPLC-ESI-MS/MS) and DNA barcoding provide insights into the chemical composition of the tested horsetail samples (Table 4). The chromatograms for each of the *E. palustre* samples were consistent with the expected HPTLC profile. The alkaloid levels in the LC-MS/MS results confirmed the identity of the *E. palustre* samples and were in accordance with the results of DNA barcoding. Chemical profiles from extracted plant materials are known to vary depending on the environmental conditions such as the geographical region in which the plant was grown and the season during which the plant material was harvested [9]. Reflecting this variation are the HPTLC chromatograms in this current study, where inter-species variation is obvious. HPTLC could successfully identify and distinguish the samples of *E. arvense* from that of *E. palustre.* However, while the European Pharmacopeia monograph proposes that the presence of two green bands at the bottom of the chromatogram is indicative of adulteration with *E. palustre*, samples of identical species (Figure 2: *E. arvense*: lanes 4, 5, 9, 11 and 12; Figure 3: *E. palustre*: lanes 4, 7, and 10) showed some striking differences in terms of bands, colors, and intensities, indicating potential difficulties in identification, and particularly with the adulteration or contamination of one species with another species. An interesting observation was that the Canadian samples (A2, A8, and A9) had a different profile in terms of the colors and bands compared to the German samples (A3–A7). Given that only the European plant specimens conformed exactly to the acceptance criteria, it calls into question whether the criteria presented by the European Pharmacopoeia can be universally applied to plant specimens from geographical locations outside of Europe. Further studies involving horsetail specimens from an array of different countries and locations would be needed to be undertaken to confirm this.

Moreover, two samples of *E. arvense*, namely A6 and A7, showed faint green bands in the region of Rf below 0.1 (Figure 2). Green bands in this region are used by the European Pharmacopeia to identify the presence of alkaloids from *E. palustre* [8], but their subjectively low intensity pardons them from such a conclusion. In fact, these green bands do not represent *Equisetum* alkaloids at all, since the LC-MS/MS analysis found no detectable *Equisetum* alkaloids in these two samples (Table 1). Conversely, the absence of green bands in this region do not guarantee that a sample is free from *E. palustre* adulteration. The HPTLC chromatogram of Sample A1 serves as a good illustration. No green bands near the application line attributed to *E. palustre* could be seen, and it could be categorized as an unadulterated sample if one depended on the HPTLC alone. However, LC-MS/MS analysis detected *Equisetum* alkaloids in A1. Therefore, it was likely contaminated by *E. palustre*. The limitations of HPTLC identification lie in the fact that this method tests for flavonoids and not alkaloids. A previous study by Saslis-Lagoudakis et al. using HPTLC to identify horsetail noted that while two alkaloid bands could be detected in the *E. palustre* samples, any alkaloids present in other *Equisetum* species were well below the detection limit of the method described by the European Pharmacopoeia. An alternative method detecting alkaloids should be developed for the monograph [2].

In the meantime, LC-MS/MS has succeeded where HPTLC has not. LC-MS/MS quantified low concentrations of alkaloids, thus allowing for accurate determination of not only the identity but also adulteration. All of the *E. palustre* samples analyzed detected much higher levels of alkaloids than the *E. arvense* samples, which in all but one of the samples, had an absence of any alkaloids above the detection limit. Though the presence of 6.2 µg/g of palustrine in sample A1 indicates possible adulteration, there has not yet been a standardized determination of toxicity to humans. Current testing simply accounts for the presence or absence of alkaloids in the *E. arvense* plant material. The monograph in the European Pharmacopoeia indicates an alkaloid band intensity of no more than 5% than is visually seen in the *E. palustre* reference chromatogram. Past studies have shown that the presence of 5% of *E. palustre* in feed is toxic to livestock [6], but further research is needed to determine the effects of such levels on humans.

In general, the success of DNA barcoding hinges upon having high quality source DNA, reducing the effects of PCR inhibitors, targeting a marker sequence that has sufficient distinguishing power between species, and using the optimal primers. It is common for different species within one genus to display very similar genetic compositions. At the same time, among the general similarities, there exist defining differences such as the identities of certain nucleotides or the total lengths of the marker. These defining differences allow for the distinguishing of one species from another. In this work, the data presented improved our understanding of *E. arvense* and *E. palustre* identification by means of DNA barcoding. A prior study has suggested that the *rbcL* marker is not effective at distinguishing between *Equisetum* species, while the *ITS2* marker failed to amplify consistently [2]. However, the data presented in this study indicates that if the correct primer pairs are used, consistent quality amplification and identification can be achieved with these two markers.

With respect to the *rbcL* marker, the *rbcL*-a primer pair consistently succeeded in amplifying the source DNA and in distinguishing between *E. arvense* and *E. palustre*. Although both species could be observed among the top 100 BLAST matches for each query, *E. arvense* was frequently listed at the topmost position for the *E. arvense* samples, and the same was true for the identification of the *E. palustre* and *E. palustre* samples. We have no doubt that this marker can be used as a reliable barcode to identify *E. arvense* and *E. palustre*. The success was most often seen with plant material that had been subject to minimal processing. Therefore, a lack of amplification using this marker could imply that the source DNA may have suffered a loss in quality or quantity; for example, due to sample processing, long storage times or other types of improper handling. Another key observation that could help explain the failure of amplification is the potential presence of PCR inhibitors. In a recently published article, Kretschmer et al. considered PCR inhibitors to be a potential cause of a less-than-expected amplification success rate [22]. This suggests that inhibitors could be a common obstacle when DNA barcoding plants. Such inhibitors can be eluted along with DNA during the extraction process. Particularly observed, often in the case of *E. arvense* samples and less so in that of *E. palustre*, amplification succeeded only when the DNA elution was diluted 10-fold. The differences in the chemical composition of the two species could explain the different amounts of PCR inhibitors present in the DNA elution. Future studies should aim at elucidating the identity of the said inhibitors as well as the minimum dilution factor necessary to circumvent PCR inhibition.

Regarding the *ITS* marker, Saslis-Lagoudakis et al. used the primer pair ITS3 + ITS4 first developed in 1990 and noted that the *ITS2* region of the *ITS* marker did not show consistent amplification [2,23]. This was also observed when a similar primer pair was used in this study: ITS-S2F + ITS4. The target region for these primer pairs covers a portion of the 5.8S rRNA gene, *ITS2*, and a portion of the 26S rRNA gene, which constitute a popular barcoding region within the *ITS* marker. To further evaluate *ITS*, a different primer pair was attempted: the ITS-p3 + ITS-u4 (p3u4) pair, newly designed and published by Cheng et al. [21]. Consistent amplification success was found with both *E. palustre* and *E. arvense* species using p3u4, which interestingly targets a similar region to the two primer pairs previously mentioned. This indicates that *ITS* is still a robust and potent marker to use for the identification of *E. palustre* or *E. arvense*, provided that the optimal primer pair is adopted. Another primer pair was also attempted: the ITS-p5 + ITS-u2 pair (p5u2), also based on Cheng et al. [21]. Unlike p3u4, p5u2 targets a different region of the ITS, covering a portion of the 18S rRNA gene, *ITS1*, and a portion of the 5.8S rRNA gene. P5u2 frequently led to good DNA sequences and does not appear to be affected by dilution in the same manner as the other primer pairs. For example, while other primers needed dilution in order to have amplification, this was not the case with p5u2. However, when the p5u2-derived sequences were queried in BLAST, they did not consistently identify *Equisetum*. In this regard, we conclude that p3u4 is the primer pair of choice when using the *ITS* marker to identify horsetail species. Future work can be conducted to further study the p5u2 primer pair and its value in the identification of horsetail.

The length of the p3u4-derived *ITS* sequence from *E. palustre* (464–473 bases) was at least 149 bases longer than that of *E. arvense* (306–315 bases). This is a major defining difference that can distinguish between the two species. Moreover, within the *ITS* region where the two species do align, single nucleotide differences can be seen (Appendix A). These nucleotide differences can also contribute to distinguishing between the species. It was also observed that even within the *E. arvense* samples, nucleotides can differ from sample to sample, as notably seen with samples A7 and A9. Whether this is a testament to intraspecific variation of this particular species, or a simple consequence of PCR quality, remains to be determined. Future efforts that utilize larger sample sizes could help to shed light on the potential presence of intraspecific variation.

It is our recommendation that both the *rbcL* marker and the *ITS* marker be utilized when barcoding horsetail species to ascertain the identification. The effectiveness of this approach was clearly demonstrated in the case of samples P7, A3, and A5, where the *rbcL* marker failed to amplify these samples, but following an alignment with p3u4-derived *ITS* sequences from the other vouchered samples, their identities were reaffirmed.

Since DNA barcoding relies on one or more reference libraries, which query tools such as BLAST compare against, there is always a need to properly enrich said library. We found that GenBank, one such library used by BLAST, possesses inadequate amounts of reference accessions for *E. arvense* and *E. palustre* in terms of the *ITS* marker. In the case of *E. palustre*, no *ITS* accessions currently exist. In the case of *E. arvense*, existing *ITS* accessions have raised questions and doubts. As previously mentioned, the genetic compositions of various species within the same genus should be largely similar. Therefore, the accuracy of the existing *E. arvense ITS* accessions in GenBank was questioned when it was noticed that they matched no *Equisetum* species other than themselves, and were even strongly matched to a species of green algae. Even more alarming was the discovery that three of the accessions, MW040322.1.1, MW040323.1, and MW040324.1, came from a publication that had been retracted on 27 October 2021 [24]. On the other hand, the p3u4-derived *ITS* sequences in our work consistently matched species within the *Equisetum* genus, and the observations were logical and adhered to expectations. With the use of quality source material such as numerous vouchered plants and HPTLC and HPLC-ESI-MS/MS analyses, we are confident in the credibility of our results in identifying these plants. Our DNA sequencing results allowed us to contribute to the improvement of GenBank by adding accurate *E. arvense* and *E. palustre ITS* accessions, which can provide reliable references for future horsetail studies. Our contributions to GenBank include the following accessions:

*Equisetum palustre*:ON713470ON713471ON713472ON713473ON713474ON713475ON713476ON713477

*Equisetum arvense*:ON713478ON713479ON713480ON713484ON713485ON876507ON876508

Recently, the combination of DNA sequencing, HPTLC fingerprinting, and HPLC-ESI-MS compound identification was also applied in discriminating Zicao samples from unknown sources. Discrimination was found to be possible with HPTLC, if two different mobile phase systems are applied. DNA sequencing, on the other hand, has the advantage of the effective species identification of every sample from which DNA barcodes could be amplified [22]. This current study confirms the value of using a combination of analytical techniques in the identification of purported samples of *Equisetum*. Highlighted is the value of LC-MS/MS in identifying adulteration, even at trace levels, and the authoritative conclusions drawn from DNA barcodes while HPTLC provides colorful snapshots of comparative chemical profiles for each sample.

In summary, our study shows that the combination of optimized DNA barcoding, HPTLC, and HPLC-ESI-MS/MS provides additional benefits in the identification of *E. arvense* and *E. palustre* samples compared to any singular method.

## 4. Materials and Methods

### 4.1. Plant Material

Nineteen *E. arvense* and *E. palustre* samples were sourced from Canada, Germany, the European Pharmacopoeia, and from commercial entities. All of the horsetail samples were given unique identification numbers upon arrival at our facility and were later assigned test codes for easy identification during analysis (Table 5). *E. arvense* sample A1 was sourced from Sunrise Botanics while *E. palustre* sample P1 was sourced from Chromadex. Both are finely milled botanical materials from commercial entities. *E. arvense* samples A2, A8, and A9 were wild-crafted horsetail from two different locations in British Columbia in the spring (A8 and A9) and summer (A2) of 2021. Samples A8 and A9 were botanically verified and vouchered (Figure 5) by our resident research scientist Jan Slama (J.S.) at the University of British Columbia Herbarium. *E. palustre* sample P2 was sourced from the European Pharmacopoeia. The remaining *E. arvense* and *E. palustre* samples A3 to A7 and P3 to P10 were wild-crafted from fields in Oberstüter and Schüllinghausen, Germany. These were designated with voucher ID codes that identify the species, time of year, and location that the plant specimen were obtained. The format of this voucher ID was alphanumerical; ‘EA02′ representing *E. arvense* specimen taken from the Schüllinghausen Feld, ‘EA05′ representing *E. arvense* specimen taken from Südstraße Haspe, ‘EP03′ representing *E. palustre* specimen taken from Wodantal, and ‘EP05′ representing *E. palustre* specimen taken from Oberstüter. Included in the code were the letters ‘F’ for spring (Frühling in German), ‘H’ for autumn (Herbst in German), and ‘S’ for summer (Sommer in German). ‘20′ or ‘21′ were added to indicate the years 2020 and 2021, respectively. Thus, a sample bearing the voucher ID ‘EP03S21′ would be understood to be an *E. palustre* sample obtained in the summer of 2021 from Wodantal.

### 4.2. HPTLC

Rutin, hyperoside, and caffeic acid were each prepared in methanol (1 mg/mL) as the reference standards. The wildcrafted horsetail samples were dried and ground to powder form prior to sample preparation. Both the *E. arvense* and *E. palustre* samples were extracted and prepared following the European Pharmacopoeia 8.0 monograph for *Equisetum* [8]. A total of 1.0 g of sample was extracted in 10 mL of methanol (Fisher Scientific Chemicals, Waltham, USA), vortexed, and sonicated in a water-bath at 60 °C for 10 min, allowed to cool, centrifuged at 3800 rpm for 5 min, and then filtered into vials. A total of 5 µL bands of 8 mm of each extracted sample solution was applied by the Automatic TLC Sampler (ATS4, CAMAG, Muttenz, Switzerland) on a HTPLC silica gel 60 F_254_ plate (Merck, Sigma, Rahway, USA). A 1:1:1 mixture of the reference standards rutin, hyperoside, and caffeic acid was sprayed onto the first track. The HPTLC plates were developed in a saturated chamber up to 70 mm with a relative humidity of 33% in the Automatic Developing Chamber (ADC2, CAMAG, Muttenz, Switzerland) using a mobile phase consisting of ethyl acetate (Fisher Scientific, Waltham, USA), distilled water, glacial acetic acid (Sigma Aldrich, Burlington, USA), and formic acid (Sigma Aldrich, Burlington, USA) (67:18:7.5:7.5 *v*/*v*/*v*/*v*). Detection was achieved by spraying the plate with 2-aminoethyl diphenylborinate (Sigma Aldrich, Burlington, USA; 10 mg/mL solution in methanol) and then by spraying with polyethylene glycol 400 (Sigma Aldrich, Burlington, USA; 50 mg/mL solution in ethanol) using the derivatizer (CAMAG, Muttenz, Switzerland). After 5 min, the plate was examined under ultraviolet light at 366 nm using a TLC Visualizer (Visualizer 2, CAMAG, Muttenz, Switzerland).

### 4.3. HPLC-ESI-MS/MS

Sample preparation was performed according to Tipke et al. and Müller et al. [5,6] with some modifications. In brief, dried plant material was ground with a Retsch (Mixer Mill MM 400, Haan, Germany) ball mill for 2 min at 30 s^−1^. Two different sample preparation schemes were applied for plant material with a low or no alkaloid content (*E. arvense* samples) and plant material with a high alkaloid content (*E. palustre* samples).

(a)Plant material with a low/no alkaloid content:

Approximately 50 mg of the powdered plant material was accurately weighed into a 2 mL Eppendorf tube. One milliliter of 1 M hydrochloric acid was added, and the mixture was extracted for 14 h in a tube rotator (Multi Bio RS-24, Biosan, Riga, Latvia) with the following settings: orbital: 21/01, reciprocal: 15/01, vibrio: 5/1. After centrifugation for 10 min at 12,100× *g*, the supernatant was collected while the pellet was again re-suspended in 0.5 mL of 1 M hydrochloric acid and re-extracted for 30 min including a final centrifugation step. Both extracts were combined and mixed with 500 µL dichloromethane. The two-phase liquid was vortexed (90 s, 1200× *g*) and centrifuged (90 s, 12,100× *g*). The organic phase was discarded and 240 µL of 25% ammonium hydroxide was added to neutralize the extract. For further purification, this solution was transferred to a 1.5 g diatomaceous earth column (Isolute HM-N, Biotage, Uppsala, Sweden). After 20 min of incubation on the column, the alkaloids were eluted with 6 x 4 mL of dichloromethane. The combined organic fraction was evaporated to dryness under a stream of nitrogen at 40 °C. Then, the obtained dry residue was carefully re-dissolved in 2 mL of HPLC-MS grade methanol and diluted 1:100 using methanol, resulting in a dilution factor of 200 compared to the calibration data.

(b)Plant material with a high alkaloid content:

Approximately 50 mg of the powdered plant material was accurately weighed into a 2 mL Eppendorf tube and extracted with 1.7 mL of methanol (80%) for 15 min in an ultrasonic bath in sweep mode (Elmasonic S120, 12.75 L, 37 KHz, 200 W, Elma Schmiedbauer GmbH, Elma, Germany). The suspension was centrifuged for 7 min at 12,100× *g*. The supernatant was pipetted and filtered, while the pellet was re-suspended another three times using 1.3 mL methanol (80%) each. All of the organic extracts were combined and evaporated to dryness at 65 °C under a stream of compressed air. The final residue was re-dissolved using 2 mL of HPLC-MS grade methanol in an ultrasonic bath for 15 min (Elmasonic S120, 12.75 L, 37 KHz, 200 W). This solution was first diluted by a factor of 100, and subsequently by a factor of 50 using HPLC grade acetonitrile (70%), resulting in a dilution factor of 10,000 compared to the calibration data.

All samples were prepared in duplicates, and in all cases, the difference between the two results were less than 15%. The means of the two duplicate measurements were presented in this study. The analysis of the *Equisetum* alkaloids was carried out as described by Tipke et al. [2] using a 1200 Series HPLC (Agilent, Waldbronn, Germany) coupled with a 3200 QTrap mass spectrometer (Sciex, Framingham, MA, USA). Electrospray ionization tandem mass spectrometry (ESI-MS/MS) detection was performed in ESI positive multiple-reaction-monitoring mode (MRM). For chromatographic separation, a 100 × 2.1 mm (1.7 μm) Poroshell 120 hydrophilic interaction liquid chromatography (HILIC) column was used (Agilent, Waldbronn, Germany) using gradient elution, as described by Tipke et al. [5].

### 4.4. DNA Barcoding

Each raw sample was weighed to 10 mg (±5 mg) for extraction. Depending on the total number of samples, either the semi-automated MagMAX™ Plant DNA Isolation Kit (SKU A32549, Thermo Fisher Scientific Inc., Waltham, MA, USA) along with a MagMAX™ Express 96 magnetic particle processor (SKU discontinued, Thermo Fisher Scientific Inc., Waltham, MA, USA), or the manual, column-based SureFood^®^ PREP Advanced Kit (SKU S1053, R-Biopharm AG, Darmstadt, Germany), was used. All extracted DNA was stored at 4 °C until use or placed in a freezer at −20 °C for long-term storage.

PCR Primers: For each sample, primer pairs for the following markers were used:*rbcL*-a: targets a segment (denoted “a”) of the *rbcL* gene [15]*ITS* (ITS-p3 & ITS-u4 pair): targets the internal transcribed spacer region; specifically, part of the 5.8S rRNA gene, *ITS2,* and part of the 26S rRNA gene [21]

The markers *rbcL*-a and *ITS* were chosen because they are commonly used barcoding regions for plants [14,15,20,21,25]. Additionally, these markers were also used by other researchers in another horsetail study [2].

For use with the commercially available BigDye™ Direct Cycle Sequencing Kit (SKU 4458687, Thermo Fisher Scientific Inc., Waltham, MA, USA), the primer sequences were modified by adding to their 5′ ends either forward or reverse “M13 tails” (see Table 6 for the complete primer sequences). Sequences of the “M13 tail” are listed in the product protocol of the commercial kit. The complete primer sequences were purchased from Integrated DNA Technologies (Coralville, USA) as a dried solid in individual tubes. Upon arrival, each tube was pulse-centrifuged with a tabletop centrifuge (SKU SC1008-M, Southwest Science, Trenton, USA), then reconstituted to 100 µM per tube with distilled water. For each target, the forward and reverse primers were combined into a cocktail, with a final concentration of 0.8 µM per primer. Then, each cocktail was added to the corresponding PCR reactions. When not in use, the 0.8 µM primer cocktails were stored at −20 °C.

The product protocols of the BigDye™ Direct Cycle Sequencing Kit and BigDye XTerminator™ Purification Kit (SKU’s 4458687 and 4376487, respectively, Thermo Fisher Scientific Inc., Waltham, MA, USA) were followed for PCR amplification, cycle sequencing, and clean up steps. Detailed run conditions can be found in the said protocols, available from the product webpages. All PCR reactions were run using the MiniAmp™ Plus Thermal Cycler (SKU A37835, Thermo Fisher Scientific Inc., Waltham, MA, USA). To vortex the reaction plates, the IKA Vortex 3 (SKU 0003340001, IKA Works GmbH & Co. KG, Staufen, Germany) was chosen. Sanger sequencing and data collection were achieved with a 4-capillary system, the SeqStudio Genetic Analyzer (Thermo Fisher Scientific Inc., Waltham, MA, USA). All accessories for the SeqStudio instrument including the cartridge, cathode buffer, and plate septa were also purchased from Thermo Fisher Scientific Inc. Sanger sequencing data were obtained in the ABI file format. Using “SeqStudioDefault” as the analysis protocol, the samples were analyzed with the Sequencing Analysis Software v6.0 (Thermo Fisher Scientific Inc., Waltham, USA). Theoretically, the best sequencing data are characterized by:Distinct and non-overlapping fluorescence signals (represented by peaks in an electropherogram).Fluorescence signal intensity between 100 and 1000 relative fluorescent units (RFUs).Quality value (QV) above 10 for mixed bases, or above 20 for pure bases.

For each marker, base calls with adequate quality (as determined by the software) were selected and queried in the online Nucleotide BLAST program (“megablast” algorithm, “nr/nt” databases). This program scans genetic databases such as “GenBank”, aligns the query sequence against DNA sequence records from different organisms, and uses an algorithm to assess the degree of similarity between the sequences [26]. In our case, the results are presented with a graphical representation of the top 100 matching sequences (matches), which shows query coverage as well as color-coded alignment scores. Additionally, BLAST generates a list of the 100 top matches, and presents various calculated parameters such as “Max Score”, “Total Score”, “Query Cover”, “E value” (Expect value), and “Per. Ident” (Percent Identity). The score is calculated by assigning positive points to each pair of matched bases in an alignment, and deducting points from any mismatches or gaps between bases [27,28]. The Max Score is the highest possible score achievable by an alignment (between the query sequence and the whole or a part of a subject sequence). Therefore, when two sequences are very similar, the match would produce a high Max Score. “Query Cover”, expressed as a percentage, describes the lengths of the two sequences being aligned, relative to each other [29]. The “E value” is a statistical value that represents the number of times that the program would find the match in the database by chance [27,28]. The match is more statistically significant when the E value is lower. “Percent Identity” shows the extent to which the two sequences of an alignment share the same bases [28]. The best match, which represents the organism species most likely to identify a sample, should have the highest “Max Score”, “Total Score”, “Query Cover”, and “Per. Ident” as well as the lowest “E value”. BLAST would place such a match at the topmost position in the list of matches. Sometimes, different species match the query sequence with equal likelihood (where all the parameters are the same). In such cases, the query can only be identified to the genus level; or, if different genera are present, it would be assessed as inconclusive.

## Figures and Tables

**Figure 1 plants-11-02562-f001:**
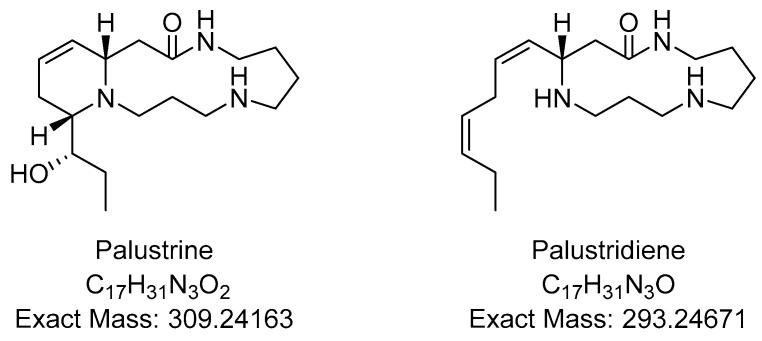
Structures of known *Equisetum* alkaloids [13].

**Figure 2 plants-11-02562-f002:**
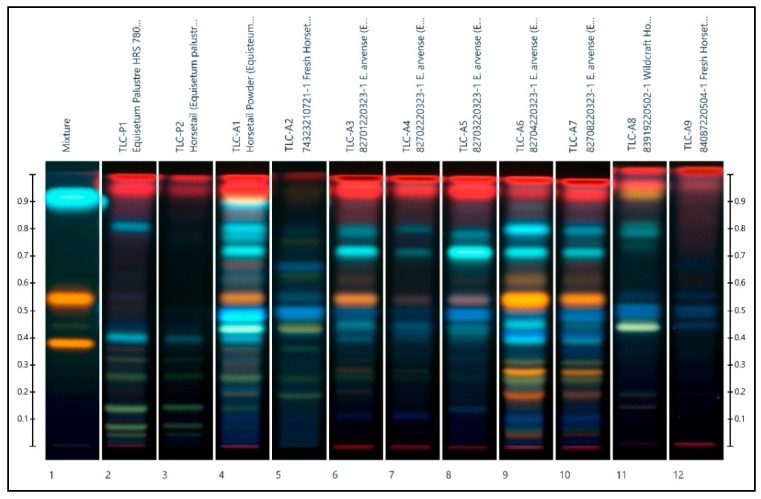
HPTLC chromatogram showing the *E. arvense* extracts of A1 to A9 (tracks 4 to 12). The *E. palustre* extracts of P1 and P2 (tracks 2 and 3, respectively) are also shown for comparison. The chemical standards rutin, hyperoside, and caffeic acid are present in ascending order on track 1.

**Figure 3 plants-11-02562-f003:**
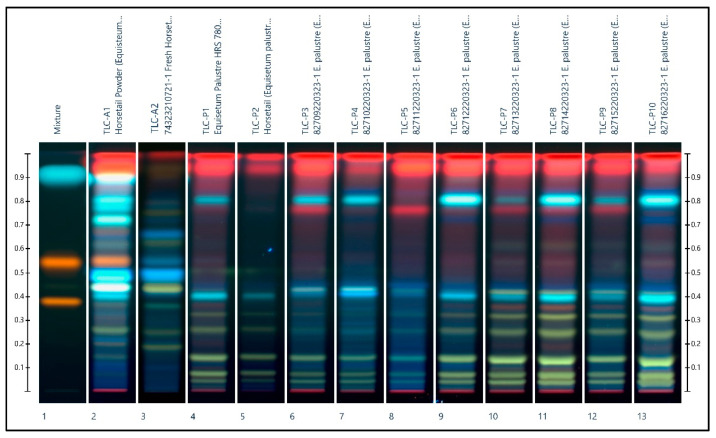
HPTLC chromatogram showing the *E. palustre* extracts of samples P1 to P10 (tracks 4 to 13). The *E. arvense* extracts samples A1 and A2 (tracks 2 and 3, respectively) are also shown for comparison. The chemical standards rutin, hyperoside, and caffeic acid are present in ascending order on track 1.

**Figure 4 plants-11-02562-f004:**
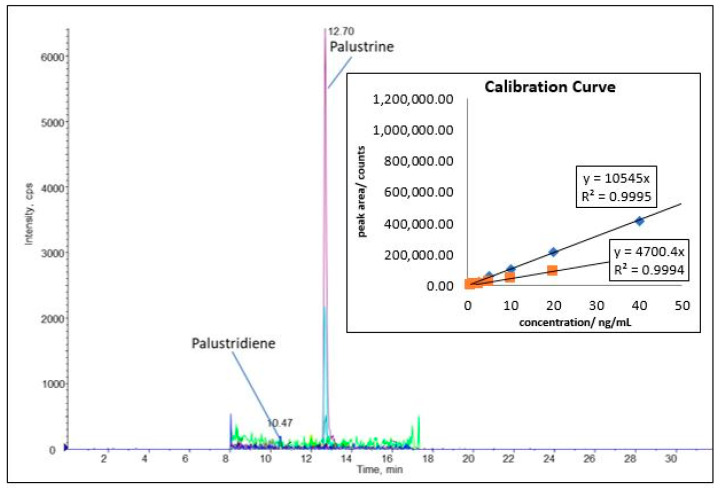
A typical HPLC-MS/MS chromatogram for *E. palustre* showing traces of palustridiene and larger amounts of palustrine. The calibration curve features the standards of palustrine (*m*/*z* 310.100 → 154.200) and palustridiene (*m*/*z* 294.000 → 157.200).

**Figure 5 plants-11-02562-f005:**
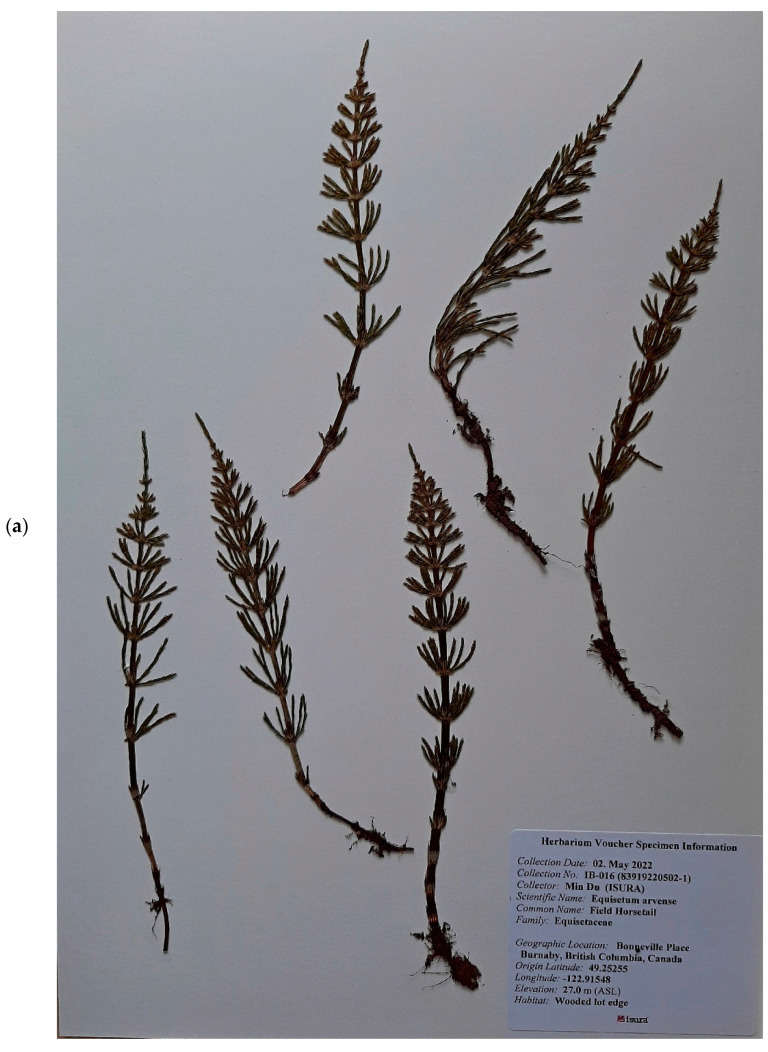
Wildcrafted *E. arvense* from (**a**) Burnaby (Test Code A8) and (**b**) Richmond (Test Code A9) in British Columbia, Canada.

**Table 1 plants-11-02562-t001:** The *Equisetum* alkaloid content of the *E. arvense* samples analyzed by LC-MSMS. Values of 0.0 indicate results below the LoD.

Test Code	Average Palustrine Content in µg/g	Average Palustridiene Content in µg/g	Sum of Alkaloids in µg/g
A1	6.2	0.0	6.2
A2	0.0	0.0	0.0
A3	0.0	0.0	0.0
A4	0.0	0.0	0.0
A5	0.0	0.0	0.0
A6	0.0	0.0	0.0
A7	0.0	0.0	0.0
A8	0.0	0.0	0.0
A9	0.0	0.0	0.0

**Table 2 plants-11-02562-t002:** The *Equisetum* alkaloid content of the *E. palustre* samples analyzed by LC-MSMS. Values of 0.0 indicate results below the LoD.

Test Code	Average Palustrine Content in µg/g	Average Palustridiene Content in µg/g	Sum of Alkaloids in µg/g
P1	769.3	38.5	807.8
P2	408.5	0.0	408.5
P3	425.9	0.0	425.9
P4	414.0	0.0	414.0
P5	572.9	0.0	572.9
P6	560.2	0.0	560.2
P7	313.2	0.0	313.2
P8	301.2	0.0	301.2
P9	469.7	0.0	469.7
P10	280.3	0.0	280.3

**Table 3 plants-11-02562-t003:** The DNA barcoding results for each horsetail sample corresponding to each gene marker. Included were the Max Score, Query Cover, E value and Per. Ident (Percent Identity) of the topmost BLAST match for each sample, using the *rbcL* marker, which frequently allowed us to distinguish between *E. palustre* and *E. arvense*. A negative result refers to samples that identified with a plant outside of the *Equisetum* genus of the horsetail family. The primer pair used for each marker is shown in parentheses. p3u4 denotes the ITS-p3 + ITS-u4 primer pair.

Test Code	*rbcL* (*rbcL*-a) (Max Score, Query Cover, E Value, Per. Ident)	*ITS* (p3u4)
P1	*E. palustre* (1216, 99%, 0.0, 99.40%)	Negative
P2	*E. palustre* (1195, 98%, 0.0, 99.39%)	*Equisetum* spp.
P3	*E. palustre* (1166, 99%, 0.0, 99.53%)	*Equisetum* spp.
P4	*E. palustre* (1201, 99%, 0.0, 99.10%)	*Equisetum* spp.
P5	*E. palustre* (1195, 100%, 0.0, 99.54%)	*Equisetum* spp.
P6	*E. palustre* (900, 78%, 0.0, 97.89%)	*Equisetum* spp.
P7	Did not amplify	*Equisetum* spp.
P8	*E. palustre* (1208, 99%, 0.0, 99.25%)	*Equisetum* spp.
P9	*E. palustre* (1086, 100%, 0.0, 99.34%)	*Equisetum* spp.
P10	*E. palustre* (996, 89%, 0.0, 96.97%)	*Equisetum* spp.
A1	Did not amplify	Did not amplify
A2	*E. arvense* (1205, 98%, 0.0, 99.10%)	*Equisetum* spp.
A3	Did not amplify	*Equisetum* spp.
A4	*Equisetum* spp. (852, 99%, 0.0, 95.91%)	*Equisetum* spp.
A5	Did not amplify	*Equisetum* spp.
A6	*E. arvense* (1138, 99%, 0.0, 97.89%)	*Equisetum* spp.
A7	*E. arvense* (1164, 99%, 0.0, 99.08%)	*Equisetum* spp.
A8	*E. arvense* (1194, 99%, 0.0, 99.54%)	*Equisetum* spp.
A9	*E. arvense* (1216, 100%, 0.0, 99.40%)	*Equisetum* spp.

**Table 4 plants-11-02562-t004:** A summary of the results gleaned from each identification method. Samples P1 and P2 were positively identified as *E. palustre* in every method. Sample A1 was revealed to have contamination and was not successfully identified as *E. arvense* with DNA. Both *rbcL* and *ITS* markers were used to conclude the identities for DNA barcoding including by cross referencing each other.

	Identification Methods		
Test Code	HPTLC	HPLC-MS/MS	DNA Barcoding
P1	*E. palustre*	*E. palustre*	*E. palustre*
P2	*E. palustre*	*E. palustre*	*E. palustre*
P3	*E. palustre*	*E. palustre*	*E. palustre*
P4	*E. palustre*	*E. palustre*	*E. palustre*
P5	*E. palustre*	*E. palustre*	*E. palustre*
P6	*E. palustre*	*E. palustre*	*E. palustre*
P7	*E. palustre*	*E. palustre*	*E. palustre*
P8	*E. palustre*	*E. palustre*	*E. palustre*
P9	*E. palustre*	*E. palustre*	*E. palustre*
P10	*E. palustre*	*E. palustre*	*E. palustre*
A1	*E. arvense*	*E. arvense*	Did Not Amplify
A2	*E. arvense*	*E. arvense*	*E. arvense*
A3	*E. arvense*	*E. arvense*	*E. arvense*
A4	*E. arvense*	*E. arvense*	*E. arvense*
A5	*E. arvense*	*E. arvense*	*E. arvense*
A6	*E. arvense*	*E. arvense*	*E. arvense*
A7	*E. arvense*	*E. arvense*	*E. arvense*
A8	*E. arvense*	*E. arvense*	*E. arvense*
A9	*E. arvense*	*E. arvense*	*E. arvense*

**Table 5 plants-11-02562-t005:** Eight horsetail samples from varying sources were obtained and allotted a test code. The alphanumeric code using “P” refers to all *E. palustre* samples, while the code using “A” refers to all *E. arvense* samples.

In-House Lot Number	Description	Source	*Equisetum* Species	Test Code
78075211104-1	Herbal reference standard	European Pharmacopoeia	*E. palustre*	P1
75890210901-1	Ground horsetail herb VBRM	ChromaDex	*E. palustre*	P2
82709220323-1	Wildcrafted horsetail, voucher ID EP03S21	Wodantal, Hattingen, Germany	*E. palustre*	P3
82710220323-1	Wildcrafted horsetail, voucher ID EP03H21	Wodantal, Hattingen, Germany	*E. palustre*	P4
82711220323-1	Wildcrafted horsetail, voucher ID EP03F21	Wodantal, Hattingen, Germany	*E. palustre*	P5
82712220323-1	Wildcrafted horsetail, voucher ID EP03H20	Wodantal, Hattingen, Germany	*E. palustre*	P6
82713220323-1	Wildcrafted horsetail, voucher ID EP05S21	Oberstüter, Germany	*E. palustre*	P7
82714220323-1	Wildcrafted horsetail, voucher ID EP05H21	Oberstüter, Germany	*E. palustre*	P8
82715220323-1	Wildcrafted horsetail, voucher ID EP05F21	Oberstüter, Germany	*E. palustre*	P9
82716220323-1	Wildcrafted horsetail, voucher ID EP05H20	Oberstüter, Germany	*E. palustre*	P10
66886210721-1	Dried horsetail herb	Sunrise Botanicals	*E. arvense*	A1
74323210721-1	Wildcrafted horsetail, not vouchered	Burnaby, British Columbia, Canada	*E. arvense*	A2
82701220323-1	Wildcrafted horsetail, voucher ID EA02S21	Schüllinghausen Feld, Germany	*E. arvense*	A3
82702220323-1	Wildcrafted horsetail, voucher ID EA02H21	Schüllinghausen Feld, Germany	*E. arvense*	A4
82703220323-1	Wildcrafted horsetail, voucher ID EA02F21	Schüllinghausen Feld, Germany	*E. arvense*	A5
82704220323-1	Wildcrafted horsetail, voucher ID EA02H20	Schüllinghausen Feld, Germany	*E. arvense*	A6
82708220323-1	Wildcrafted horsetail, voucher ID EA05H20	Südstraße Haspe, Germany	*E. arvense*	A7
83919220502-1	Wildcrafted horsetail, voucher ID IB-016 (83919220502-1), UBC Herbarium accession number V252538	Burnaby, British Columbia, Canada	*E. arvense*	A8
84087220504-1	Wildcrafted horsetail, voucher ID IB-017 (84087220504-1), UBC Herbarium accession number V252539	Richmond, British Columbia, Canada	*E. arvense*	A9

**Table 6 plants-11-02562-t006:** Complete primer sequences for the DNA-based identification of field horsetail (*Equisetum arvense*). Each sequence is a chimera of the target-specific primer sequence, connected to a M13 tail at its 5′ end. In this table, M13 tails are bolded and italicized.

Targets	Complete Sequences (5 → 3′)
*rbcL*-a	***TGTAAAACGACGGCCAGT***ATGTCACCACAAACAGAGACTAAAGC
	***CAGGAAACAGCTATGACC***CTTCTGCTACAAATAAGAATCGATCTC
*ITS* (“ITS-p3” and “ITS-u4” [13])	***TGTAAAACGACGGCCAGT***YGACTCTCGGCAACGGATA
	***CAGGAAACAGCTATGACC***RGTTTCTTTTCCTCCGCTTA

## Data Availability

Data are contained within the article and Appendix A.

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
