# Peer review of "A Multi-Pronged Technique for Identifying *Equisetum palustre* and *Equisetum arvense*—Combining HPTLC, HPLC-ESI-MS/MS and Optimized DNA Barcoding Techniques"

_plants, 2022, doi:10.3390/plants11192562_

Round 1

Reviewer 1 Report (Previous Reviewer 1)

From my point of view, this improved version meets the conditions for publication.

Author Response

Dear Reviewer,

Thank you for your response. We appreciate you taking the time to review our work.

Reviewer 2 Report (Previous Reviewer 2)

I request authors to present the DNA barcoding data empirically rather than describing the results. While authors analysing their sequence in BLAST analyses, what was the query coverage and what was the percentage of identity in BLAST? I suggest authors do carefully read DNA barcoding literature on barcoding analysis to improve their presentation of DNA barcoding data in the article. 

Author Response

Dear Reviewer,

Thank you very much for your input and suggestions regarding our manuscript. We want to improve our content as you have suggested, but would like to please seek additional clarification.

In the comments, you indicated that empirical data should be included. One way we plan on doing this is to include additional supplementary figures, where we show the actual BLAST results in full, including parameters such as “Max Score”, “Query Cover”, “E value”, etc. Along with this approach, we can add in-text content to explain the BLAST terms in more detail. Would this be sufficient, or would you prefer to have us include, in the main body of the manuscript, a sample figure (perhaps a truncated portion of a BLAST result as a representative)? Or would you have alternative suggestions for us?

Furthermore, you indicated the following areas as “must be improved”:

  • Introduction
  • Methods
  • Results
  • Conclusions

May you please advise if the revisions needed for these 4 sections relate to the aforementioned point about empirical data presentation? Or would you have further details that you can share with us on the particular deficiencies that we need to address?

Reviewer 3 Report (Previous Reviewer 3)

After the first revision, the manuscript improves consistently. I have only one question: you said that rbcL marker is able to distinguish Equisetum species but you didn't show the sequence obtained (or the blast alignment). 

Can you add the obtained sequence to the supplementary material? It would be interesting to see it (and maybe you should deposit also this data besides  ITS sequences).

Author Response

Dear Reviewer,

Thank you for your response. We appreciate you taking the time to review our work.

As requested, we have added the BLAST alignments for the rbcL marker to the supplementary material. They are shown as Supplementary Figures S19 to S33.

Round 2

Reviewer 2 Report (Previous Reviewer 2)

Article has improved a lot, I appreciate and thank the authors for their efforts to improve the article.

This manuscript is a resubmission of an earlier submission. The following is a list of the peer review reports and author responses from that submission.

Round 1

Reviewer 1 Report

The manuscript titled “A Novel ITS Sequence for the Identification of Equisetum arvense” shows the results obtained for the identification of different samples of Equisetum, considering that E. arvense species can be adulterated with a more toxic species, E. palustre, so an accurate identification is important and needed. For this purpose, the authors used three identification methods: a method from European Pharmacopoeia - HPTLC analysis and two more modern methods - HPLC-ESI-MS/MS and DNA barcoding.

The manuscript is interesting, well organized, clearly written, the experiments are appropriate and performed to a high technical standard.

I remarked that the results are clearly, and realist presented, and the conclusions are supported by the data.

I think that the informations provided are valuable. However, there are a few points that can be improved:

  1. Because the composition of samples depends on time of year in which it were collected, stage of development,environmental conditions, for the samples harvested please provide this information.
  2. For LC-MSMS analysis the results should include the statistical analysis of data.
  3. The paragraph of “discussions” should include several comparisons with other papers on the same species, comparatively discussing the results obtained. So this paragraph can be improved.
  4. Maybe it possible to use more references from the last 5 years, because only a third of them are after 2018.
  5. Please read the paper carefully for English language style and accuracy.
  6. The resolution of figures can be improved.

Reviewer 2 Report

The research article entitled “A Novel ITS Sequence for the Identification of Equisetum arvense” deals with chemical and DNA based identification of Equisetum species. Though the study is interesting, yet it needs to improved a lot for the suitability of publication.

Comments

  • The title of the article “A Novel ITS Sequence for the Identification…….”, is not completely reflecting this research study. Also, there is no “Novel ITS sequence” per se. Yet, authors claiming an ITS sequence as novel needs to be changed.
  • Methodology: One of the major concerns is voucher specimen for the collected samples. Authors needs to understand that deposition of the collected plants samples in any of the herbaria with voucher number is one among the basic criteria in DNA barcoding; without such voucher specimen, it is difficult to identify the specimen by other researchers in case if it needs clarity in the identification in the future. Therefore, authors should deposit their study species in herbaria or in raw drug herbarium. Authors may refer these article https://doi.org/10.1111/j.1471-8286.2007.01678.x and https://doi.org/10.7554/eLife.68264 .The importance of voucher specimens for phytochemical research are highlighted here https://doi.org/10.1016/j.phytochem.2017.04.004
  • Figure 4: The current Linnaeus system for identification of plants is majorly by floral characters. The provided picture neither provides the diagnostic identifying morphological characters of horsetail from other species nor a display of complete morphological characters that aids in the identification.
  • Authors should submit the generated DNA sequences in the study either in BOLD or Genbank, NCBI and should provide the accession number of those sequences in the article. Without such accession number, it is difficult to validate the claims.
  • Figure 3: Authors can provide this as supplementary figure, and authors needs to understand several factors contribute to the poor quality electropherograms. It is no surprise that Equisetum species in this study has the same issue. However, Did the authors tried to sequence the sample in replicates? One of the fundamentals of DNA barcoding is collecting the same species in distinct geographic locations and sequence the DNA barcodes. In this regard authors have sequence multiple accession and see whether the problem of poor quality electropherograms persists in all the accession or not.
  • I do not understand what the authors wants to convey by providing the nucleotides sequences in the research article. Either they can construct a phylogeny and show the dendogram or they can provide inter and intraspecies genetic distance which could help to chose the best marker to distinguish the Equisetum species

Reviewer 3 Report

This study wanted to test three identification methods (HPTLC, HPLC-ESI-MS / MS and DNA barcoding) for identifying the fraud of horsetail and discover a wrong sumbission for E.arvense ITS in GeneBank. It can be interesting to compare both chemical and biomolecular technologies but I think that using so few samples is not enough for a scientific publication but only for setting the methodology. Furthermore, for E. arvense the authors have DNA sequence only for one sample and it can't be enough for a scientific publication (considering also the lack of accession in genebank for this species). I suggest major revision, adding more samples, and trying to test and compare the three methodologies on more products, focusing on commercial items. 

Line 78: gene must be written in italic

Line 222: I would prefer to see all the three sequences aligned in one one figure

Line 297: Please add a reference

-Please add more reference in all the text, in particular in the introduction and in the discussion sections.

-I don't think that the title you chose is the most explicative for this study, because with this title you focus the attention only on ITS marker and DNA barcoding. I suggest to change it.

-In the introduction section I would add something about the regulation about this species in the food supplement sector

-How do you explain the bad quality sequence of P2?

-I think the discussion section is a little bit confusing. Try to finish the paper with a resume of the three technologies tested and highlight pro and cons of each one.

-Do you think there is one technology that is better than the other? If so, why? Do you think it is possible to use only one considering the data of your study or is it necessary to use more than one? It would be interesting to write these things in the text

-